# A Systematic Investigation on the Influence of Intumescent Flame Retardants on the Properties of Ethylene Vinyl Acetate (EVA)/Liner Low Density Polyethylene (LLDPE) Blends

**DOI:** 10.3390/molecules28031023

**Published:** 2023-01-19

**Authors:** Eid M. Alosime, Ahmed A. Basfar

**Affiliations:** 1King Abdulaziz City for Science and Technology, P.O. Box 6086, Riyadh 11442, Saudi Arabia; 2Mechanical Engineering Department, College of Engineering, King Saud University, P.O. Box 800, Riyadh 11421, Saudi Arabia

**Keywords:** intumescent flame retardant, cable and wire, EVA/LLDPE blend, extrusion, mechanical properties

## Abstract

Because of their high filler loadings, commercial-grade clean flame-retardant materials have unstable mechanical properties. To address this issue, intumescent polymers can be used to develop clean flame retardants with very low levels of smoke and toxicity generation. An intumescent flame retardant (IFR) system composed of red phosphorus (RP), zinc borate (ZB), and a terpolymer of ethylene, butyl acrylate, and maleic anhydride (EBM) was used to prepare EVA (ethylene-vinyl acetate) and EVA/LLDPE (linear low-density polyethylene) composites; their mechanical and flammability properties were systematically investigated. The limiting oxygen index (LOI) of the EVA/LLDPE (as base material) composite containing RP and ZB mixed with nonhalogenated flame retardant, mainly magnesium hydroxide (MH) and coadditives, including processing aids and thermal stabilizers, was established. RP was found to have little effect on the tensile properties of EVA/LLDPE 118W/120 phr flame-retardant (MH + RP) composites. There was a minute difference in the effective trend of RP between tensile strength and elongation at break. Following the addition of ZB, the elongation at break of the composites gradually decreased with increasing RP content and then leveled off when the RP content was over 10 phr. Mechanical properties (elongation at break and tensile strength) can be best maintained at below 10 phr content of RP. The mechanical properties decreased with lower amounts of EBM content. In addition, flame retardancy increased when the EBM content decreased. The findings further revealed that MH and RP have poor compatibility, yielding poor mechanical properties. The LOI greatly increased with RP content, even though the total content of flame retardants (main + intumescent flame retardant) was the same in all formulations. Only over 5 phr RP content formulations passed V-0 of the UL-94 test. When under 5 phr, the RP content formulations did not pass V-0 of the UL-94 test.

## 1. Introduction

A power cable is a vital part of transmitting energy and is closely related to system security [1]. Because of complex working environments, flammable insulating materials, and operating conditions, fire accidents involving power cables are common. A study found that 50% of electrical fires are caused by burning cables [2]. When a fire breaks out, it disrupts power transmission, which not only causes a large amount of economic damage, but also affects people’s health and safety [3]. Because electrical cables are made of polymeric components such as a sheath, bedding, and insulation, which contain fuel sources, they may be more susceptible to fire. Because of arcing, excessive ohmic heating (without arcing), and external heating, the fire source is responsible for starting and spreading the fire [3,4,5]. Blends comprising two materials—ethylene-vinyl acetate (EVA) polyethylene (EVA-PE) and EVA copolymer blends—are some of the most widely used polymers used for sheathing and the insulation of electric cables.

Intumescent flame retardants (IFRs) have become a topic of discussion among many researchers because they are free of halogen, do not generate dioxins during combustion, and have low toxicity and smoke production.

Generally, the IFR system is made up of three parts: the acidic source, a carbonization agent, and a blowing agent [6]. The intumescent system creates a char layer that protects the substrate from heat and oxygen by relying on decomposition induced by heat. In halogen-free flame-retardant (HFFR) polymers, metal hydroxides, particularly magnesium hydroxide (MH), have been extensively utilized as smoke- and toxin-free additives [7,8]. MH can be used in a wider range of thermoplastics because of its high decomposition temperature [9,10]. Combined with a few synergistic HFFR agents, such as red phosphorus (RP), expandable graphite (EG), and so forth, MH could lower the loading level of MH while increasing its flame retardancy. According to few studies [11,12], EG is a very effective HFFR additive that works well with other inorganic fillers. As a novel intumescent additive, RP has several drawbacks, including its propensity to spontaneously ignite and tendency to absorb moisture in the air. Masterbatches and encapsulation have been successfully implemented to resolve issues with handling safety and stability. RP is made of an amorphous inorganic macromolecular structure that relies on a condensed phase process: liquid acid films. The acids that form the film are metaphosphoric, phosphoric, and polyphosphoric acids; these films form the outer part of the burning material’s surface. In addition, the process of condensed phase forms part of the acid-promoted dehydration that causes the acceleration of the consolidated char layer formation [13,14]. Several systems, such as metallic hydroxide, halogen antimony, and phosphorous nitrogen, possess synergistic effects when used in the flame-retardant polymer [15,16].

An additional promising approach to more efficiently diminish the fire and environmental risks of these materials involves using nanoparticles, along with HFFR additives [17,18]. For instance, the fire retardancy of EVA and LDPE can be blended together using organoclay, along with one of two aluminum trihydroxides (ATH); in addition, MH has been evaluated by thermogravimetric analysis and cone calorimetric measurements for assessing the influence of the surface layer created during pyrolysis of the polymer nanocomposites by numerical models [19]. The presence of 68 wt.% of metal hydroxides considerably reduced the heat release rate (HRR); this influence was further enhanced by coupling the hydroxide with 5 wt.% organoclay. Sanchez-Olivares et al. [20] have pointed out that, in addition to the excellent fire safety and improved mechanical properties of EVA/LLDPE, the processing conditions have an important influence on the composite structure. However, the formulations consisting of 120 parts per hundred resin (phr) of ATH yielded a quite dense structure; the presence of triazine (TZR) formed porous structures with a pore number and distribution proportional to the TRZ amount. This can be attributed to the release of volatiles by TRZ, here aiding in enhancing the heat shielding properties of the produced protective layer, which has an advantage compared with the reinforcing influence of clay [21].

In the present work, various formulations containing blends of EVA/LLDPE as a base material and IFRs such as RP and zinc borate (ZB) were mixed with nonhalogenated flame retardant, mainly MH and coadditives, including processing aids and thermal stabilizers, to achieve an improvement in flammability, as well as retaining mechanical and electrical properties, of EVA/ LLDPE compounds for wire and cable applications. Efforts were made to improve the melt strength of HFFR compounds by utilizing the adhesive force of random terpolymers of ethylene, butyl acrylate, and maleic anhydride (EBM) between the base polymers and flame retardants. In addition, ethylene alpha-olefin was used as a base polymer because of its high flame-retardant load ability, which can increase the flame retardancy.

## 2. Results and Discussion

The selection of the proper main flame retardants and quantity, along with picking out the proper intumescent flame retardants and quantity, is the most important aspect of HFFR formulations. More studies on intumescent flame retardants in thermoplastic HFFR formulations are ongoing [22,23]. It has also been established that the combination could be a better initiative because it leads to poor compatibility. For instance, the combination of MH and RP has poor compatibility and, thus, produces poor mechanical properties. Similarly, RP is the master batch of PE-based concentrated RP. To enhance the capabilities of the intumescent flame, retardant formulations are made. For instance, the ZB content formulation was combined with the RP content formulation. These combinations are significant in preventing deterioration of the mechanical properties of the flame and in increasing the flame retardancy.

These findings reveal that the specific polymer influences the content and type of flame retardant required to achieve a specific purpose. The type and content of the flame retardant can affect the efficiency of the flame-retardant efficiency of the polymer. In addition, RP-based flame retardants are the best alternatives to toxic halogen additives. RP has excellent flame retardancy and is compatible with many flame retardants because of its low toxicity and synergistic effects. Thus, RP flame retardants have become popular worldwide because of their excellent features. Studies of intumescent flame retardants in thermoplastic HFFR formulations were conducted based on three varying aspects—(1) influence of RP content without ZB; (2) influence of RP content with ZB; and (3) influence of EBM content formulation—were compounded by an internal mixer, and cable extrusions were successfully conducted.

### 2.1. Influence of RP Content without ZB

RP has a pure elemental activity that allows matches to ignite, making it the most powerful flame retardant known. To assess the effect of RP without ZB on the mechanical properties of EVA/LLDPE, tensile strength and elongation at break were measured. The effect of RP content on the tensile properties of EVA/LLDPE composites is shown in Figure 1.

Tensile strength has been shown to increase with an increasing content of MH [24]. Thus, an increase in the tensile strength because of additional MH is linear over a range of 90 to 120 phr [22]. Similarly, elongation at break decreased with an increase in MH. However, the detrimental influence on mechanical properties linked to the high flame-retardant loading required for the cable application of EVA and EVA blends is a well-known issue [25]. In accordance with several studies [22,23], the current results also indicate that the deleterious effects of the flame retardant are mainly induced by the MH content.

Figure 1 shows that RP has little effect on the tensile properties of EVA/LLDPE 118 W/120 phr flame-retardant (MH + RP) composites. At the same time, there is also a slight difference in the effective trend of RP between tensile strength and elongation at break. These findings resonate with a study conducted by Chen et al. [26], who found that RP has little effect on the mechanical properties of the flame-retardant polypropylene (PP). This explains why the RP only slightly affected elongation at break and tensile strength. These effects also align with the findings of Savas et al. [27] that RP causes an increase in tensile strength by approximately 60% in thermoplastic polyurethane. This can be attributed to the use of EVA content, which has been found to enhance tensile strength by 9.97% [28,29]. EVA can be used to lower the plastic matrix’s modulus of elasticity because of its excellent flexibility and softening point. It further improves the bonding of the poplar HDPE composites, which then enhances the mechanical properties. 

With the addition of RP instead of partial ZB, tensile strength almost levels off. However, elongation at break of the composites gradually decreases with increasing RP content and then levels off when the RP content is over 10 phr. The reason is not clear at the moment, but it is conceivable that this could occur because the lower content of RP acts as an interfacial additive and then enhances the interfacial adhesion between the filler and polymer matrix. This implies that the change is because of RP rather than ZB because when the latter was added alone, the tensile strength leveled off, which is an indication that there would be no effect without it [30].

This aligns with Liang et al. [31], who stated that ZP acts as a synergist, thus boosting the tensile strength and elongation at break of the composites. However, the findings differ from those of Chen et al. [26]: ZB was found to enhance tensile strength and elongation at break. Although the tensile strength of the composites levels off when ZB is used, the combinatory effects show that they are compatible, although this has some limits [29]. Chen et al. [26] established that combining additives with a compatibilizer significantly enhances tensile strength and modulus. However, at high RP content formulations, even the total content of flame retardants (main + intumescent flame retardant) are the same in all the formulations, and the mechanical properties slightly decrease with the use of RP. It is proposed that the compatibility of MH and RP is poor, resulting in poor mechanical properties. Therefore, under 10 phr content of RP is required for maintaining the mechanical properties of PP/MH/RP. These findings reveal that good compatibilization between the MH and RP requires good reactivity toward the amide and the intermediate solubility inherent in the polymer material [32]. In addition, the concentration of the compatibilizer in the composite should be sufficient to successfully incorporate the fibers in the polymer matrix while not being too high to cause disturbances with the matrix polymer phase and to lower the mechanical properties, such as strength. For these reasons, formulations of low-content RP without ZB were conducted, as shown in Table 1. The first group (AP-587, AP-662, AP-663, AP-664, and AP-665) of 80% EVA/20% LLDPE 110W examined the effects of different contents of flame retardants (magnesium hydroxide and red phosphorus) without zinc borate, with the combined contents fixed at 120 phr.

Table 1 lists the UL-94 ratings and LOI test results for the EVA/LLDPE composite samples without the ZB and MH + RP additives. As shown in Figure 2, the LOI greatly increased with an increasing amount of RP content, even though the total content of flame retardants (main + intumescent flame retardant) was the same in all formulations. These findings are similar to the results of Feng et al. [33], who found that increasing the flame-retardant content leads to an increase in the LOI. However, only content comprising formulations greater than 5 phr RP pass V-0 of the UL-94 test. Under 5 phr, the RP content formulations cannot pass V-0 of the UL-94 test. This indicates that 5 phr RP is the minimum to pass V-0 of the UL-94 test when only RP is formulated as an intumescent flame retardant. Smaller RP content formulations and a combination of ZB formulations are recommended to avoid a decrease in mechanical properties and to increase flame retardancy. In addition, their synergistic effects lead to better mechanical properties. Thus, mixing magnesium with other additives is highly recommended, as long as it produces synergistic effects. Lee et al. [34] established that adding high levels of MH filler affects the mechanical properties of composite materials.

MH influences the flame retardancy and mechanical properties of the composite polymer material, namely the EVA/MH, via the LOI. Other factors that can be influenced include tensile tests, which as shown in the previous work, have a synergistic effect [34,35]. MH, especially when prepared in an alkaline environment, improves the retardancy of the EVA polymer. These findings concluded that the developed IFR systems allow for a reduction of the RP loading down to 25% with improved mechanical properties while granting IFR performance suitable for electrical cable applications. The present work provides a viable solution for the preparation of IFR EVA-PE blends with reduced costs and improved efficiency [20]. Thus, the developed formulations have acceptable volume resistivity, as required for cable insulation material.

The morphology of the prepared composites was studied using scanning electron microscopy (SEM). Appendix A displays SEM micrographs of the EVA/LLDPE 118W and selected corresponding composites containing 120 phr flame retardants (MH + RP). 

### 2.2. Influences of RP Content on ZB-One

Smaller RP content formulations and a combination of ZB formulations are suggested to avoid a decrease in mechanical properties and increase in the flame retardancy. The influences of ZP content on ZB in EVA/EXACT 8201/CLNA-8400/EBM/120 phr flame-retardant (MH+ZB) formulations are shown in Table 2. The second group (AP-651, AP-670, AP-671, AP-672, and AP 673) of 75% EVA/10% EXACT 8201/6% CLNA-8400/9% EBM examined the effects of adding various contents of red phosphorus at a fixed magnesium hydroxide and zinc borate at 120 phr.

It is considered that EXACT 8201 acts as an intermediate material between EVA/CLNA-8400 and EBM/120 phr MH. If EXACT 8201 does not cooperate with these materials, tensile strength may decrease with EXACT 8201 content. From these results, a 10% content of EXACT 8201 in the base polymer can be used in EVA/CLNA-8400/EBM/120 phr flame-retardant (MH) formulations (Appendix A).

As shown in Figure 3, the mechanical properties do not largely change with increased RP content when in the range of 2–5 phr. However, they undergo a slight change. The previous study showed that RP content was necessary to increase flame retardancy in the range of nondecreasing mechanical properties. In addition, as shown in Figure 4, under 2 phr, the RP content formulation did not pass V-0 of the UL-94 test. Only formulations with over 2 phr RP content passed V-0 of the UL-94 test. It is suggested that at least 2 phr RP content with ZB can pass V-0 of the UL-94 test in EVA/EXACT 8201/CLNA-8400/EBM/120 phr flame-retardant (MH+ZB) formulations. It has been established that MH can effectively lower flammability by approximately 50% in polymer composites. MH was used with other combinations, such as with ZB and boric acid [36]. However, these composites have been found to have mechanical properties with a marginal reduction within the added flame retardants. The flame retardant is not expected to affect the physical properties of the final product; however, it may be able to enhance the material both chemically and physically so that it has better mechanical properties. Based on the nature of the additives, they can act chemically or physically in the solid, liquid, or gas phase. For instance, halogenated compounds are said to function mostly by a vapor-phase flame-hindering mechanism over a radical reaction, while phosphorous compounds diminish the creation of flammable carbon comprising gases by increasing the conversion of polymeric materials to a char residue during pyrolysis [37].

The results of the mechanical properties and flame retardancy have shown that the range of 2–5 phr RP content with 6 phr ZB was suitable for meeting mechanical and flame-retardant properties. Among the above formulations, AP-672 and AP-673 met most of the target values, and AP-673 was selected as the basic formulation for the next formulation study. 

In addition, confusing results were observed from the cone calorimeter test, as shown in Figure 5. The tendency between RP content, PHRR time, and total HRR is not shown. Namely, the non-RP content formulation (AP-651) and 5 phr RP content formulation (AP-673) indicated almost the same PHRR time and total HRR, whereas the 3 phr RP content formulation (AP-671) showed higher and quicker PHRR. These results may come from experimental errors or unknown factors. In general, a higher RP content formulation showed lower PHRR; however, AP-671 (3 phr RP content formulation) showed the highest PHRR, exhibiting the lowest FPI. Such a correlation between FPI and the time to flashover of a substantial fire was observed with a decrease in the FPI, advancing the time to flashover [38]. In this case, the ability of low RP content was probably because of promoting the formation of a more efficient barrier to volatiles, and heat transfer allowed for further reducing the FPI values. However, definite differences were observed from residuals after the cone calorimeter (see Figure 6). Tar formation was observed in RP content formulations, and tar formation from RP can increase flame retardancy. 

### 2.3. Influence of RP Content on ZB-Two

The base polymer study found that Tafmer DF805 was a very good intermediate between EVA and LLDPE. Similar to EXACT 8201 formulations, to avoid a decrease in the mechanical properties and to increase the flame retardancy, smaller RP content formulations and a combination of ZB formulations were suggested in this study. In a previous study of CLNA-8400 (AP-654 to AP-657) in base polymer formulations, it was found that the suitable content of Tafmer DF805 in the base polymer was 15% (details of a range of properties of EVA/LLDPE-based HFFR compounds as a function of Tafmer DF805 content are given in the Appendix A). Therefore, 15% Tafmer DF805 in the base polymer was applied for all formulations. In another study, a combination of 8 wt% RP and 7 wt% ZB led to an increase in the LOI by 27.3 vol% and a UL-94 rating (V-0) [39]. The study found that combining the two flame retardants could enhance the char barrier formation and lower the mass loss rate, hence improving the flame retardancy. The study established a strong conviction that RP and ZB produce a strong synergistic effect that can improve flame polymers’ wear performance and retardance [39]. The influences of RP content on ZB in EVA/Tafmer DF805/CLNA-8400/EBM/120 phr flame-retardant (MH+ZB) formulations are shown in Table 3. The third group (AP-656, AP-674, AP-675, AP-676, and AP-677) of 70% EVA/15% Tafmer DF805/6% CLNA-8400/9% EBM examined the effects of adding various contents of red phosphorus at a fixed magnesium hydroxide and zinc borate at 120 phr.

As shown in Figure 7, the mechanical properties did not largely change with the increase in RP content at the range of 2–5 phr. These results are almost the same as the EXACT 8201 formulations. The previous study showed that RP content was necessary to increase flame retardancy in a range of nondecreasing mechanical properties. RP has been used as a flame-retardant additive in various polymers [39]. In addition, as shown in Figure 8, under 2 phr, the RP content formulation does not pass V-0 of the UL-94 test. Only over 2 phr RP content formulations could pass V-0 of the UL-94 test. It is suggested that at least 2 phr RP content with ZB can pass V-0 of UL-94 test EVA/Tafmer DF805/CLNA-8400/EBM/120 phr flame-retardant (MH+ZB) formulations. These results are also the same as the EXACT 8201 formulations. The results of the mechanical properties and flame retardancy showed that the range of 2–5 phr RP content with 6 phr ZB was suitable for meeting the mechanical and flame-retardant properties. Among the above formulations, AP-677 was selected as the basic formulation for the next formulation study.

In addition, interesting results were observed from the cone calorimeter test, as shown in Figure 9 and Figure 10. The tendency between RP content, PHRR time, and total heat release rate was observed. Namely, non-RP content formulation (AP-656) showed a higher PHRR than RP content formulations (AP-675 and AP-677). In addition, as shown in Figure 9, a higher LOI formulation showed a lower PHRR. A higher RP content formulation generally showed higher LOI and lower peak heat release rates. It was found that RP increased flame retardancy in the LOI and cone calorimeter tests. In addition, as in previous results of residuals after the cone calorimeter, tar formation was observed in RP content formulations (see Figure 11). As described, tar formation from RP increased the flame retardancy. The two most important mechanical properties of a flame retardant are impact strength and tensile strength [40]. However, for the present study, the focus was on the latter. In general, there are some instances in which the tensile strength varies with the content of the retardant. An example is the unsaturated polyester, where the tensile strength was found to increase as the ammonium polyphosphate (APP) loading increased. In their study, Yue et al. [40] found that a decrease in the mechanical properties, specifically the tensile strength, occurred because of the poor wetting or adhesion with the polymer matrix. The materials were, however, found to be less than 15 phr, which is lower than 20% of their mechanical properties. Thus, the percentage of 19 MPa tensile strength and an LOI of 26.2% make for excellent flame retardancy.

### 2.4. Reducing EBM Content in EVA/EXACT 8201/CLNA08400 Formulations—One

In a previous study of EBM (AP-638 to AP-641), as given in Appendix A, it was clear that EBM formulations showed higher mechanical properties compared with LLDPE 118 W formulations in all different formulations. In the case of tensile strength, LLDPE 118 W formulations showed around 12 MPa, and EBM formulations showed around 14 MPa. Moreover, in the case of elongation at break, the values of EBM formulations were much higher (over 190%) than those of LLDPE 118 W formulations (under 190%). In addition, after thermal aging, EBM formulations also presented higher elongation at break in all different contents. Different contents exhibited varying degrees of influence on elongation at break and tensile strength. Each content introduced leads to unique properties, leading to an enhancement of the flame properties. This explains the change in the properties of the flame retardants. It was considered that the adhesive strength by EBM between the base polymer and flame retardants increases the mechanical properties of HFFR formulations. However, it was found that the flame retardancy of the EBM formulations was much lower than that of the LLDPE formulations. EBM formulations presented 6–8% lower LOI than LLDPE in all different content formulations. Here, butyl acrylate and maleic anhydride in EBM caused reduced flame retardancy. For the above reasons, reducing the EBM content in EVA/EXAT 8201/CLNA08400 formulations is suggested for increasing flame retardancy, as shown in Table 4. AP-673 was selected as the basic formulation for this purpose. The changes and impacts on the mechanical properties of the flame retarded formulations reiterate what the literature has established: high levels of flame-retardant additives significantly affect the polymer matrix and, as such, lower their mechanical performance [41]. It is this problem that EBM may be able to address. Thus, the content of the retardants must be manageable so as to affect the polymer matrix because this would consequently affect the mechanical properties. Even if there are sufficient compounded conditions, an increase in the content of flame retardants can decrease the mechanical properties [22]. The fourth group (AP-673, AP-686, AP-687, AP-688, and AP-689) examined the effects of adding various contents of EBM into 75% EVA/10% EXACT 8201/6% CLNA-8400 at fixed magnesium hydroxide, zinc borate, and red phosphorus at 125 phr.

As shown in Figure 12, as estimated, the mechanical properties (elongation at break and tensile strength) decreased with the EBM content. It is already known that EBM contributes to increasing the mechanical properties; therefore, EBM is used to enhance such properties. Accordingly, it is possible to conjecture the optimum EBM content to meet the proposed mechanical properties. As shown in the figure, even the 1 phr EBM content formulation was able to meet the mechanical properties of the IEC 60502 specification, whereas it did not meet our target value. In addition, as shown in Figure 13, the flame retardancy increased with a decrease in EBM content. All formulations met V-0 of the UL-94 test. This result coincides with the estimation from a previous study [40]. For instance, the LOI value of neat unsaturated polyester increased to 28.7% from 24.8%, but after adding the APP content, it increased to 25 from 10 phr, an indication that APP is a good flame retardant. Thus, an increase in the phr can indicate that the flame retardant is excellent. However, it was further established that the increase was because of the presence of phosphorous and nitrogen, which are inherent in APP. These two additives have synergistic effects that lead to an increase in the LOI of the retardant [42]. Therefore, from the mechanical and flame retardancy results, a suitable formulation can be selected for use in wire and cable applications because all formulations meet the specifications. HFFR materials are required to meet electrical properties, mechanical properties, flame retardancy, and low smoke generation. The thermal aging testing conditions of HFFR insulation materials for using halogen flame retardant insulation materials in wire and cable applications is 100 °C for 168 h. Moreover, the cone calorimeter test showed interesting results, as shown in Figure 14 and Figure 15. It was observed that a higher EBM content formulation (AP-673) showed lower LOI and PHRR, while a lower EBM content formulation (AP-689) showed higher LOI and PHRR. The same conflicting results are observed in EBM content formulations when looking at the relationship between the LOI and PHRR. Higher flame retardancy generally means a higher LOI value and lower PHRR. However, in EBM content formulations, this relationship has yet to be elucidated. For these unique results, more research is needed; it is proposed that EBM influences the increasing levels of flame retardancy, even showing a lower LOI value. In addition, strong tar formation was observed in the EBM content formulations (see Figure 16). As described, it is apparent that tar formation comes from EBM, which increases flame retardancy. EBM also increases the bonding power (adhesive strength) between polymers and inorganic flame retardants. Therefore, tar formation becomes strong with increased EBM content. 

As shown in Figure 17, volume resistivity (Ωcm) decreased with increased EBM content in the base polymers. The electrical property suddenly decreased by a small amount of EBM. AP-688 (3% EBM content in base polymers) showed 3.1 × 10^14^ (Ωcm), while AP-689 (1% EBM content in base polymers) showed 1.0 × 10^15^ (Ωcm). It is believed that decreased electrical properties can be caused by acrylate (CH_2_ = CHCOOR) and maleic anhydride (cis-HO_2_CCH = CHCO_2_H) in EBM. Therefore, the suitable content of EBM should be studied in formulations for insulation purposes. However, in jacket formulations, a maximum of 9% EBM content in the base polymers can be applied because a volume resistivity of 7.3 × 10^13^ (Ωcm) does not make a problem in practical cable application.

### 2.5. Reducing EBM Content in EVA/EXAT 8201/CLNA08400 Formulations—Two

Because the results of reducing the EBM content in EVA/EXAT 8201/CLNA08400 formulations are very important, all formulations can be used in actual products, and it is necessary to reconfirm the results. Similar formulations of reducing EBM content in EVA/EXAT 8201/CLNA08400 for increasing flame retardancy have been conducted, as shown in Table 5. The fifth group (AP-730, AP-731, AP-732, AP-733, and AP-734) examined the effects of adding various contents of EBM and CLNA-8400 into 75% EVA/10% EXACT 8201 at fixed magnesium hydroxide, zinc borate, and red phosphorus at 125 phr.

The same results were observed from the second trial formulations, reconfirming that all the results of the first trial formulations (AP-686 to AP-689) were correct. Figure 18 shows that the mechanical properties (both elongation at break and tensile strength) decreased with decreasing EBM content. However, all formulations met the mechanical properties of the specifications of IEC 60502. This differs from previous studies that have shown that mechanical properties can decrease with the flame retardants’ increasing content, despite sufficient compounded conditions [22]. High-retardant materials and EVA/LDPE are some of the more widely used base polymers. High flame retardants are widely used because of their load ability, which increases the flame retardancy. The major flame retardants are inorganic materials, such as HH, MH, and aluminum trihydroxide (ATH), which have high decomposition temperatures and smoke-suppressing abilities. A heavy loading of flame-retardant materials of more than 50% *w/w* is needed to achieve high flame retardancy. However, there is a need to be careful because heavy loading of flame retardants leads to a deterioration of the mechanical properties/shape memory effects [22]. Thus, maintaining the level of the flame retardants to standard levels is essential in preventing the mechanical properties from being compromised.

In addition, as shown in Figure 19, flame retardancy increased with decreasing EBM content. All the formulations met V-0 of the UL-94 test. This result coincides with a previous study [22]. From the mechanical and flame retardancy results, a suitable formulation can be selected for the final purpose because all the formulations met the specifications.

As shown in Figure 20, volume resistivity (Ωcm) decreased with increasing EBM content in the base polymers. These results are almost the same as in the previous study. That is, the electrical property decreased suddenly by a small amount of EBM. AP-731 (4% EBM content in base polymers) showed 1.6 × 10^14^ (Ωcm), while AP-730 (2% EBM content in base polymers) showed 8.3 × 10^15^ (Ωcm). However, in jacket formulations, a maximum of 9% EBM content in the base polymers can be applied. These findings align with previous studies’ results that a polymer’s electrical property depends on its size [43]. As a result, a larger size yields low electrical properties. Thus, the electrical properties decreased when the size/volume was small. However, when the size was large, the electrical properties were high. As a result, a small-sized polymer will have a low electrical property—specifically electrical resistivity—while a large-sized polymer will display higher volume resistivity [44]. The findings indicate that the polymer’s electric resistivity depends on the film’s thickness [45]. The thickness and size become important when selecting a polymer because of its electric volume resistivity. Volume resistivity is an electrical insulation property.

## 3. Experimental Section

### 3.1. Materials

Evaflex 360 (ethylene vinyl acetate, vinyl acetate content: 25%, melt mass-flow rate (MFR): 190 °C/2.16 kg: 2.0 g/10 min, producer: DuPont-Mitsui Polychemicals Co., Tokyo, Japan), LLDPE 118 W (liner low-density polyethylene, melt flow index (MFI): 1.0 g/10 min, producer: SABIC., Riyadh, Saudi Arabia), CLNA-8400 (linear low-density polyethylene, MFI: 0.7 g/10 min, Hanwha Wire & Cable Compound, Seol, Korea), EBM (ethylene-butyl acrylate-maleic anhydride terpolymers, Lotader 3210, MFI: 5.0 g/10 min, ARKEMA, Colombes, France), EXACT 8201 (Octene-1 Plastomer, MFR: 190 °C/2.16 kg: 1.1 g/10 min, ExxonMobil Chemical Company, Houston, TX, USA), and Tafmer DF805 (ethylene alpha-olefin, MFR: 190 °C/2.16 kg: 0.5 g/10 min, Mitsui Chemical Inc., Shiodome, Japan) were used as the base polymers. MH (magnesium hydroxide, MAGNIFIN A Grades H10A, Mg(OH)2 content: ≥99.8%, particle size, sieve residue, >45 µm: ≤0.1%, Albemarle, Paris, France) was used as the flame retardant because of its high decomposition temperature and smoke suppression capabilities. RP (red phosphorus masterbatch, Exolit RP 692, phosphorus content: approx. 50% (*w/w*), Clariant, Cergy, France) and ZB (zinc borate, Firebrake ZB, melting point: phase change at 650 °C, Borax, Boron, CA, USA) were used as secondary flame retardants. Naugard Q (polymerized 1,2-dihydro-2,2,4-trimethylquinoline, melting range: 85–105 °C, Uniroyal Chemical, London, UK) was used as an antioxidant. Carbon black (Corax N550, ash content: 0.5%, iodine adsorption: 43 mg/g, Degussa, Munich, Germany) was used as the coloring agent.

All materials were of a commercial grade and did not go through further purification.

### 3.2. Methods

Polymer matrix pellets (EVA, LLDPE 118 W, EXACT 8201, CLNA-8400, EBM, and Tafmer DF805) were melted and mixed in an internal mixer 350S (Brabender Co., Duisburg, Germany) at a speed of 40 rpm for 4 min at the following temperatures (according to the method described by Basfar et al. [22,46]):

EVA: 120 °C, EVA/LLDPE 118W, EVA/EXACT 8201/CLNA-8400/EBM, and EVA/Tafmer DF805/CLNA-8400/EBM: 150 °C. The EVA to LLDPE 118 W, and EXACT 8201/CLNA-8400/EBM. Tafmer weight ratios were kept varied within the ratios of 80:20, 75:25, and 70:30.

Then, the rest of the additives, including the flame retardants (120–125 phr), antioxidant (1.5 phr), and coloring agent (6 phr), were mixed with matrix polymers for 10 min at 150 °C. The premixed compounds were moved to a two-roll mill (Brabender Co., Duisburg, Germany) for fine blending. The temperature of the two-roll mixer was kept around 150 °C, and the mixture was processed for 5–10 min. The mixture was moved to a hot press and compressed at 165 °C for 20 min. Sheets of the test specimens were prepared with dimensions of 110 × 185 mm and a thickness of 2 mm. Figure 21 shows a schematic diagram of the preparation method of the composites.

### 3.3. Characterizations

Mechanical properties (tensile strength and elongation at break) were measured using a universal testing machine Model 5543 from Instron, Norwood, MA, USA, in accordance with ASTM D 638M and with the following testing conditions: speed of 50 mm/min at 25 °C. Thermal aging of the samples was performed at 100 °C for 168 h with a heat-aging oven, here in accordance with IEC 60811-1-2.

The flammability of the prepared formulations was characterized by UL-94 and the limiting oxygen index (LOI) flammability tests, as well as by the cone calorimeter. UL-94 flammability tests were performed using a flammability chamber from CEAST Co., Torino, Italy, in accordance with ASTM D635 for the horizontal position and ASTM D3801 for the vertical position. The LOI test was performed using an apparatus from Fire Testing Technology Limited (Incorporating Stanton Redcroft), London, UK, in accordance with ISO 4589 (ASTM D2863). The LOI corresponds to the minimum percentage of oxygen needed for the combustion of specimens (80 × 10 × 1 mm) in an oxygen–nitrogen atmosphere. A cone calorimeter from Fire Testing Technology Limited (Incorporating Stanton Redcroft), london, UK, was used to measure heat release, in accordance with ASTM 1354-04a under a heat flux of 50 kW/m^2^, which corresponds to the heat involved during a fire. The peak heat release rate (PHRR) is considered to be the parameter best expressing the maximum intensity of a fire, indicating the rate and extent of fire spread. The time to ignition (TTI) and fire performance index (FPI), which are defined as the ratio of TTI to PHRR, provide a parameter related to the time available to escape in a real fire situation.

Volume resistivity was measured at room temperature (25 °C), here in accordance with ASTM D257, using a high resistance meter of Model HP4339B, HP, USA.

Scanning electron micrographs of the selected blended formulations were analyzed using a Jeol scanning electron microscope (Model JSM 5800, Tokyo, Japan). Specimens were cryogenically fractured in liquid nitrogen and then sputter coated with a conductive layer.

## 4. Conclusions

(1)Influence of RP content without ZB

At the low RP content formulations, mechanical properties are almost constant despite increasing RP content. However, at the high RP content formulations, mechanical properties slightly decrease by RP. It is proposed that the compatibility of MH and RP is poor, resulting in poor mechanical properties. LOI greatly increases with increases in RP content even when total content of flame retardants (main + intumescent flame retardant) is the same in all formulations. However, under 5 phr RP content formulations cannot pass V-0 of the UL-94 test. It is supposed that 5 phr RP content is the minimum to pass V-0 of the UL-94 test when only RP is formulated as the intumescent flame retardant. For avoiding decreases in the mechanical properties and for increasing flame retardancy, smaller RP content formulations and combinations of ZB formulations are recommended.

(2)Influences of RP Content on ZB

RP and ZB were found to influence the mechanical properties of the flame-retardant composites. The current study found that at least 2 phr of the combined content of RP and ZB can pass V-0 of the UL-94 test in EVA/EXACT 8201/CLNA-8400/EBM/120 phr flame-retardant (MH + ZB) formulations. In addition, the findings led to the conclusion that the suitable rate for meeting the mechanical and flame-retardant properties was a range of 2 to 5 phr of RP content with 6 phr of ZB.

(3)Reducing EBM Content in EVA/EXAT 8201/CLNA08400 Formulations

Two mechanical properties—tensile strength and elongation at break—were tested and were found to have an indirect relationship with EBM content. Thus, the content increased elongation at break while tensile strength decreased and vice versa.

Another key indirect relationship was between flame retardancy and EBM content, which increased as the other decreased. This is true compared with what the literature has established: high flame-retardant additives significantly affect the polymer matrix and, as such, lower their mechanical performance. It is this problem that EBM tries to solve. Thus, the retardants’ content must be manageable to affect the polymer matrix because this would consequently affect the mechanical properties. Even if there are sufficient compounded conditions, an increase in the content of the flame retardants can decrease the mechanical properties.

## Figures and Tables

**Figure 1 molecules-28-01023-f001:**
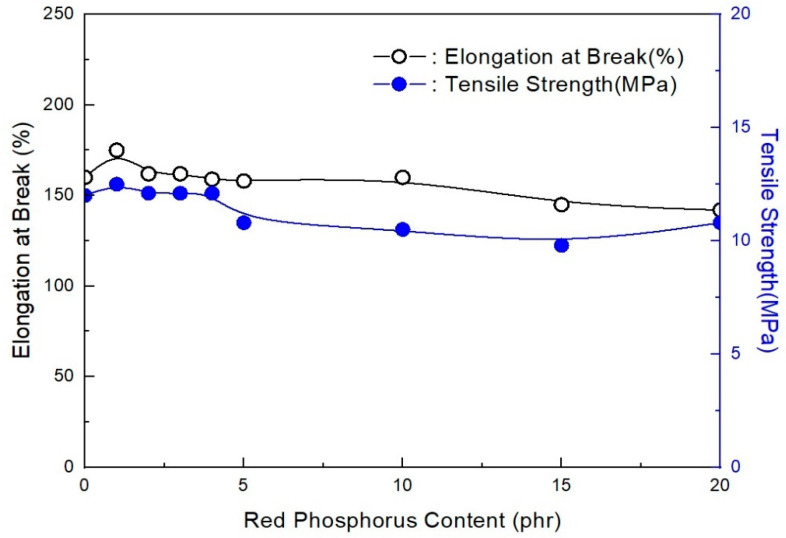
Mechanical properties of EVA/LLDPE 118W/120 phr flame-retardant (MH + RP) formulations as a function of RP content.

**Figure 2 molecules-28-01023-f002:**
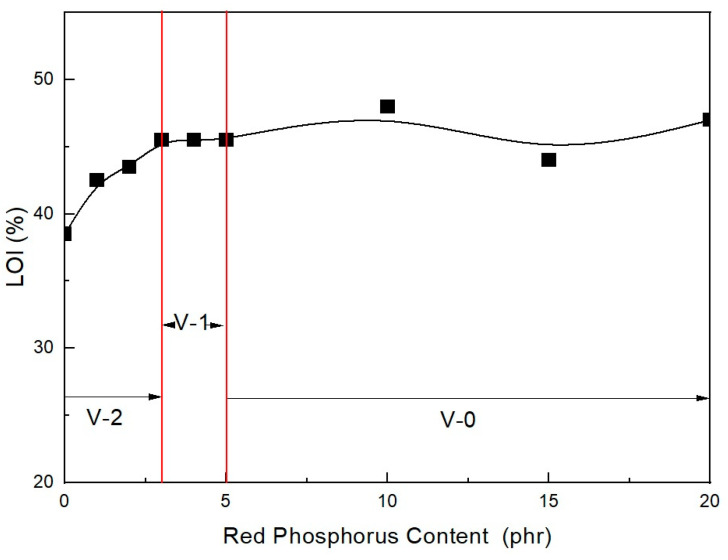
Flame retardancy of EVA/LLDPE 118W/120 phr flame-retardant (MH + RP) formulations as a function of RP content.

**Figure 3 molecules-28-01023-f003:**
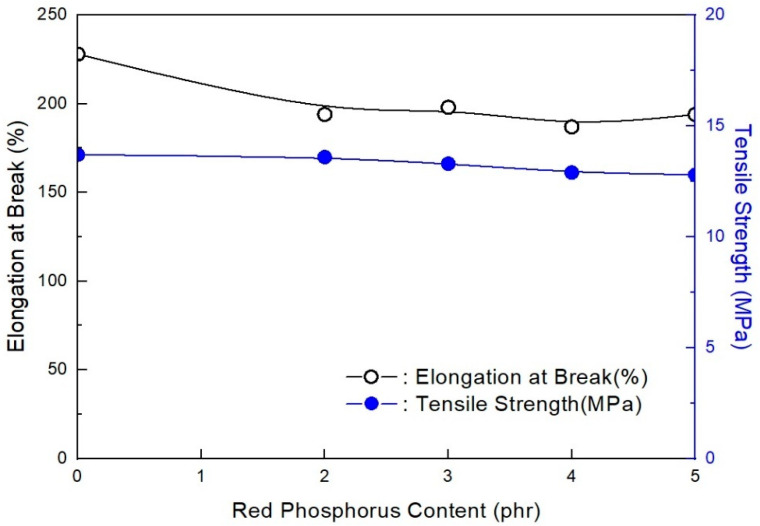
Mechanical properties of EVA/EXACT 8201/CLNA-8400/EBM/120 phr flame-retardant (MH+ZB) formulations as a function of additional RP content.

**Figure 4 molecules-28-01023-f004:**
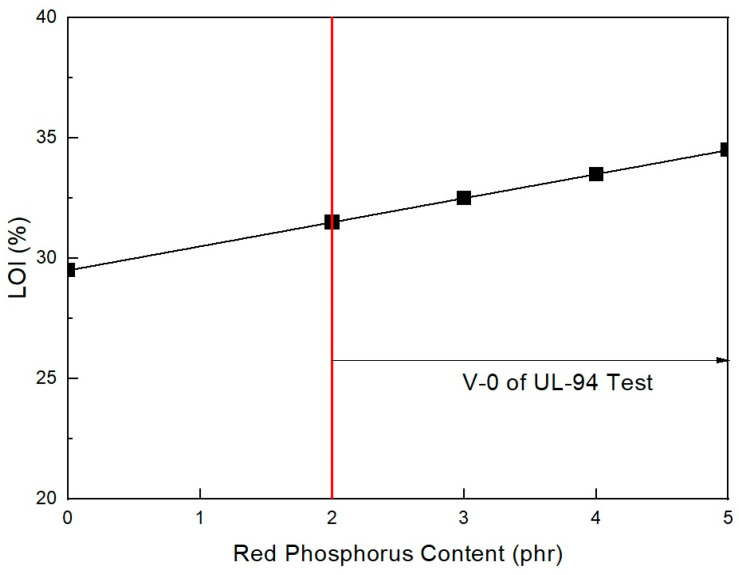
Flame retardancy of EVA/EXACT 8201/CLNA-8400/EBM/120 phr flame-retardant (MH+ZB) formulations as a function of additional RP content.

**Figure 5 molecules-28-01023-f005:**
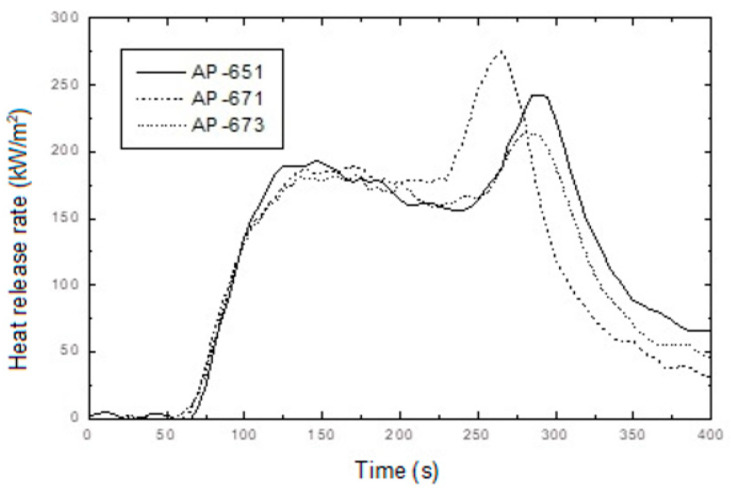
Heat release rate curves of AP-651, AP-671 and AP-673 as a function of burning time.

**Figure 6 molecules-28-01023-f006:**
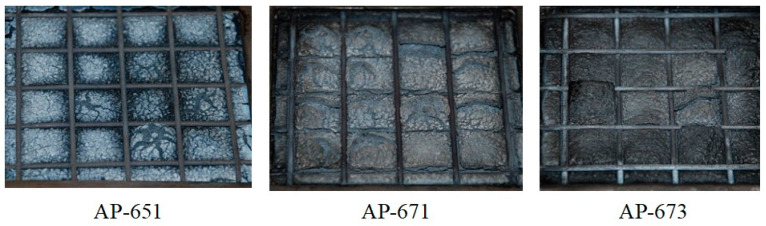
Photographs of the residua after cone calorimeter test of AP-651, AP-671 and AP-673.

**Figure 7 molecules-28-01023-f007:**
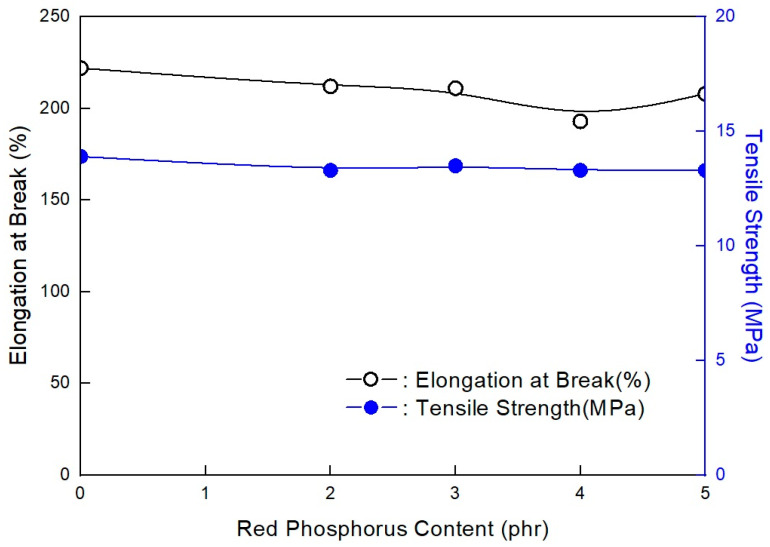
Mechanical properties of EVA/Tafmer DF805/CLNA-8400/EBM/120 phr flame-retardant (MH+ZB) formulations as a function of additional RP content.

**Figure 8 molecules-28-01023-f008:**
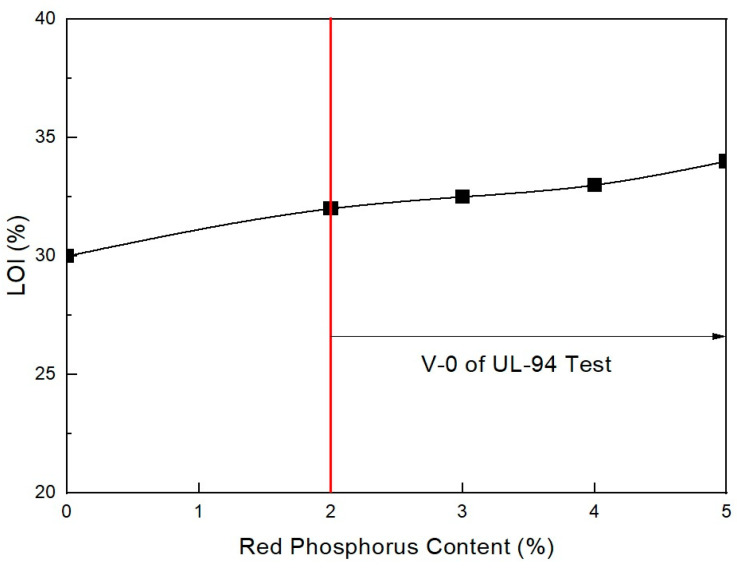
Flame retardancy of EVA/Tafmer DF805/CLNA-8400/EBM/120 phr flame-retardant (MH+ZB) formulations as a function of additional RP content.

**Figure 9 molecules-28-01023-f009:**
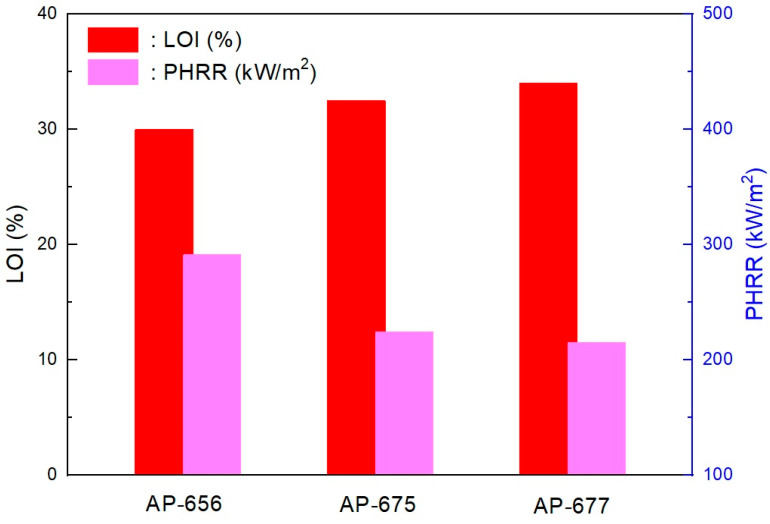
Relationships between LOI and PHRR in RP content formulations. RP content: AP-656 = 0, AP-675 = 3 phr and AP-677 = 5 phr.

**Figure 10 molecules-28-01023-f010:**
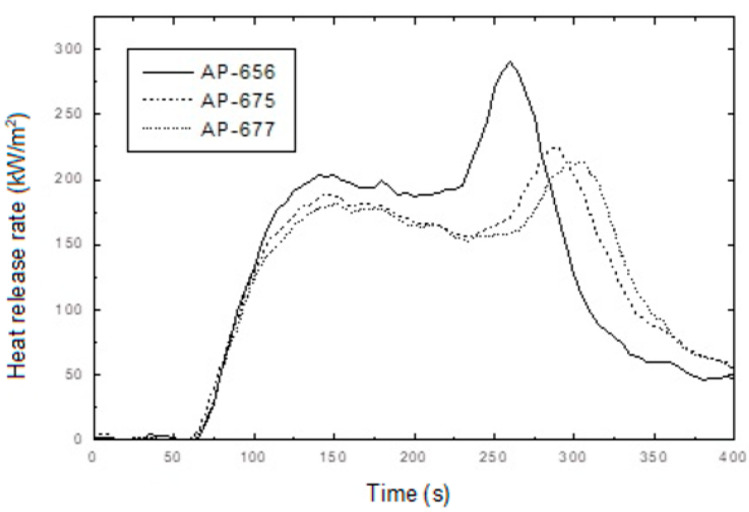
Heat release rate curves of AP-656, AP-675 and AP-677 as a function of burning time.

**Figure 11 molecules-28-01023-f011:**
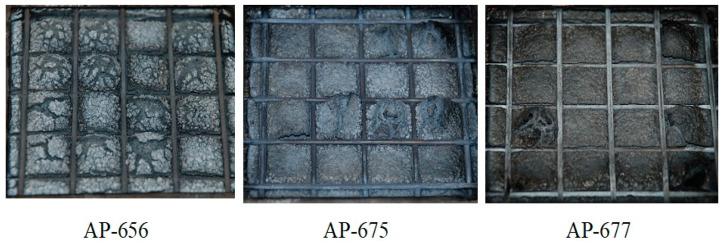
Photographs of the residua after cone calorimeter test of AP-656, AP-675 and AP-677.

**Figure 12 molecules-28-01023-f012:**
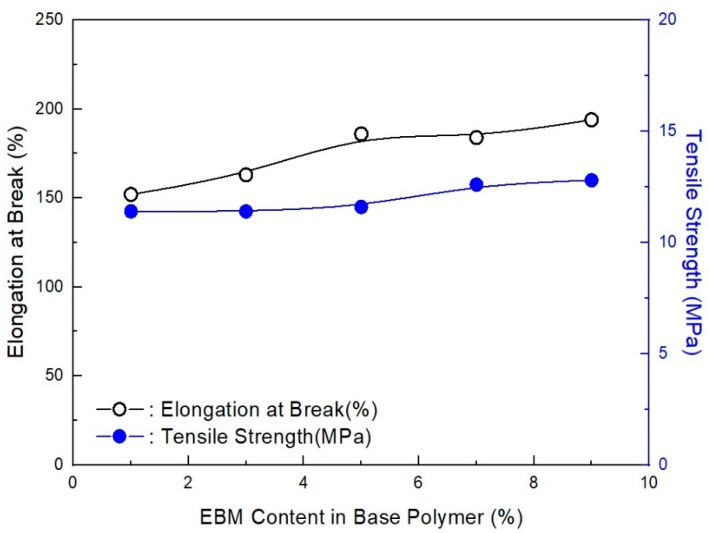
Tensile strength and elongation at break of EVA/EXACT 8201/CLNA-8400/. 125 phr flame-retardant (MH+ZB+RP) formulations as a function of EBM content.

**Figure 13 molecules-28-01023-f013:**
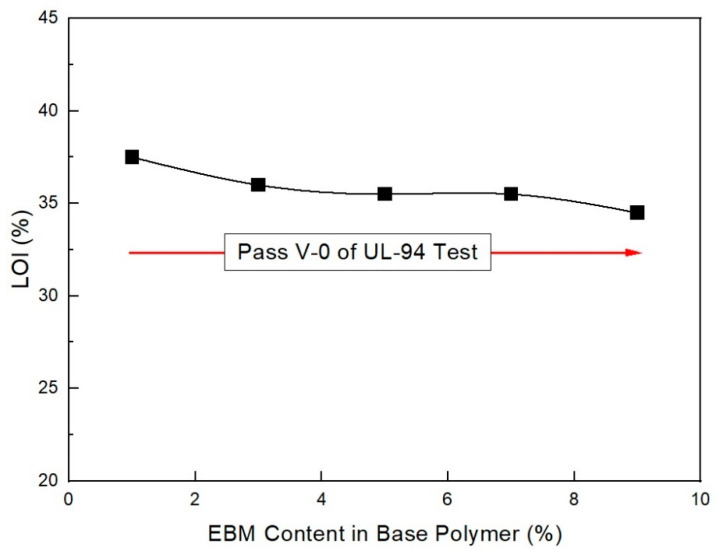
LOI (%) of EVA/EXACT 8201/CLNA-8400/. 125 phr flame-retardant (MH+ZB+RP) formulations as a function of EBM content.

**Figure 14 molecules-28-01023-f014:**
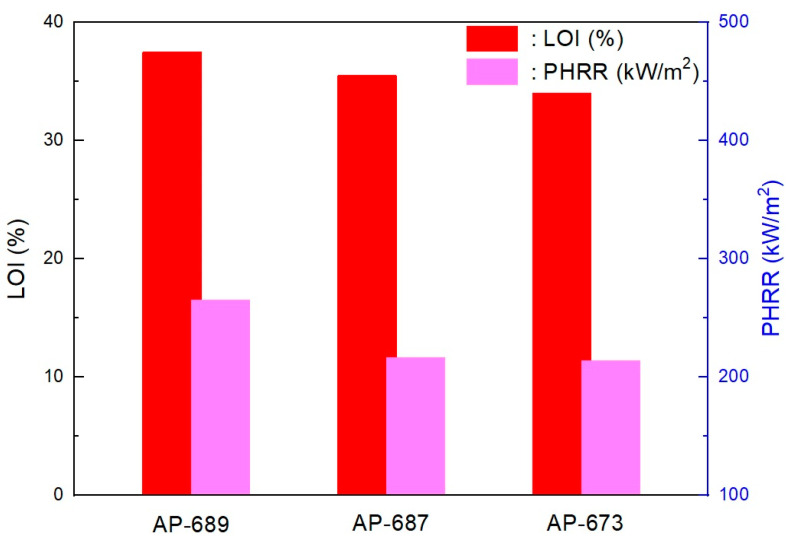
Relationships between LOI and PHRR in EBM content formulations. EBM content: AP-689 = 1 phr, AP-687 = 5 phr and AP-689 = 9 phr.

**Figure 15 molecules-28-01023-f015:**
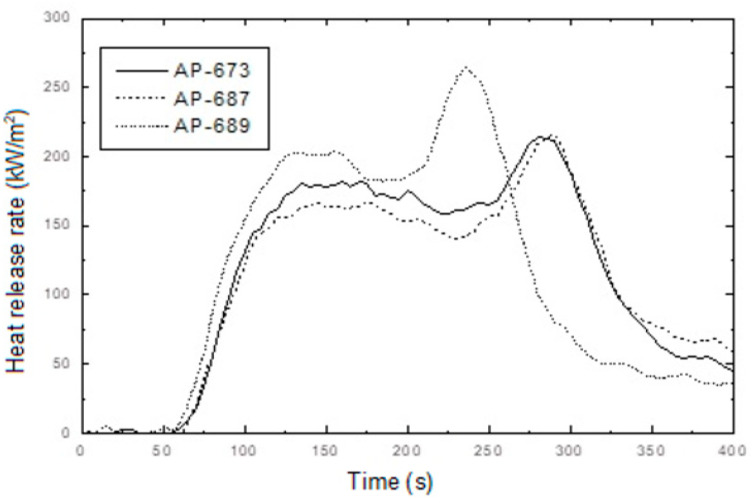
Heat release rate curves of AP-673, AP-678 and AP-689 as a function of burning time.

**Figure 16 molecules-28-01023-f016:**
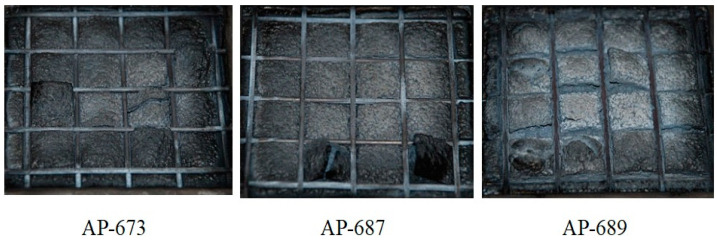
Photographs of the residua after cone calorimeter test of AP-673, AP-678 and AP-689.

**Figure 17 molecules-28-01023-f017:**
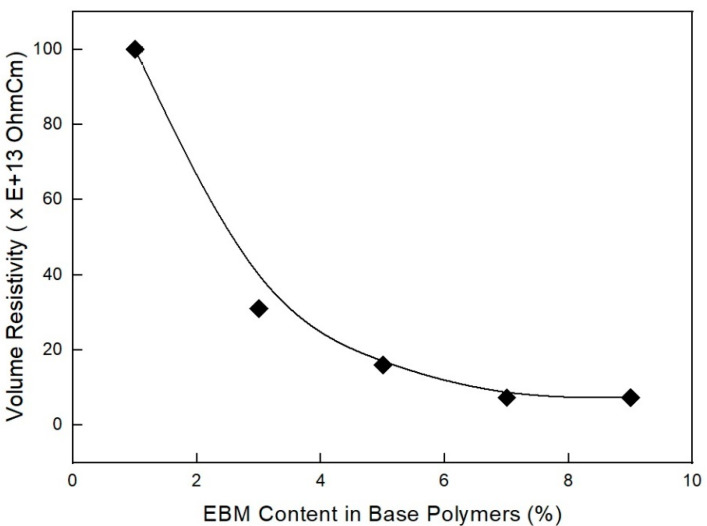
Volume resistivity (Ωcm) of EVA/EXACT 8201/CLNA-8400/. 125 phr flame-retardant (MH+ZB+RP) formulations as a function of EBM content.

**Figure 18 molecules-28-01023-f018:**
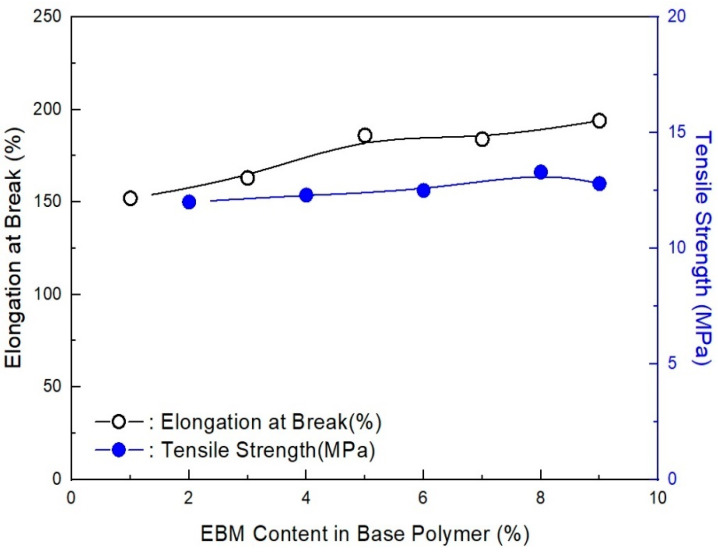
Tensile strength and elongation at break of EVA/EXACT 8201/CLNA-8400/125 phr flame-retardant (MH+ZB+RP) formulations as a function of EBM content.

**Figure 19 molecules-28-01023-f019:**
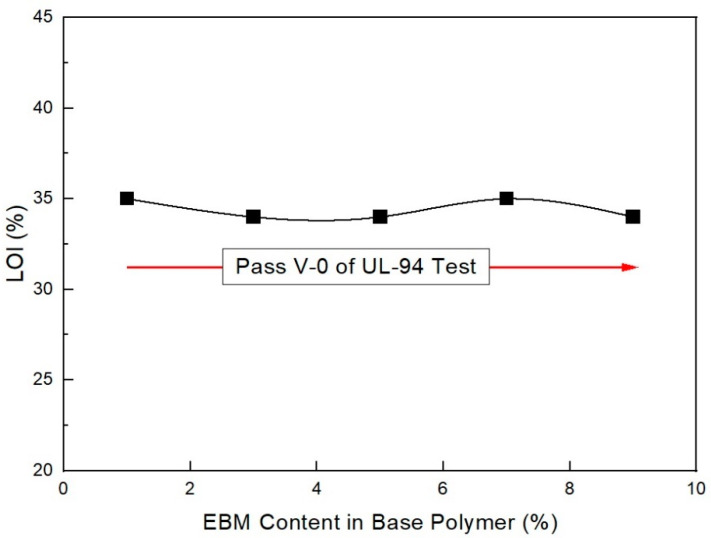
Flame retardancy of EVA/EXACT 8201/CLNA-8400/125 phr flame-retardant (MH+ZB+RP) formulations as a function of EBM content.

**Figure 20 molecules-28-01023-f020:**
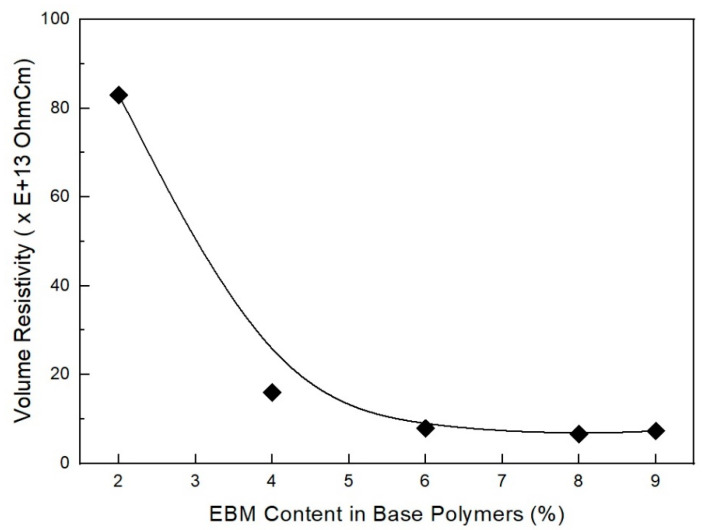
Volume resistivity (Ωcm) of EVA/EXACT 8201/CLNA-8400/125 phr flame-retardant (MH+ZB+RP) formulations as a function of EBM content.

**Figure 21 molecules-28-01023-f021:**
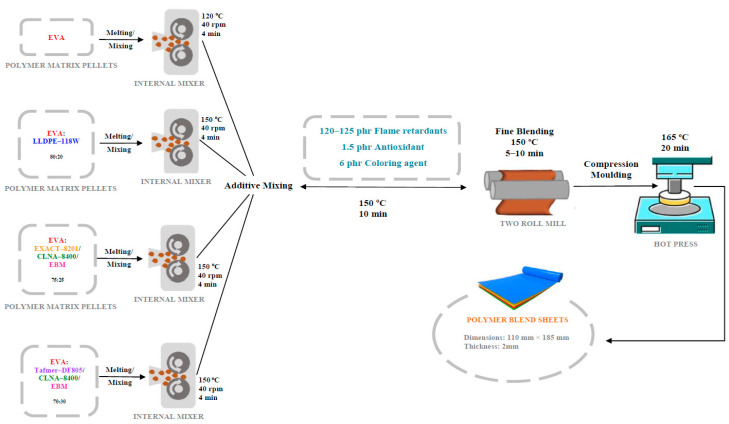
The schematic diagram of the preparation method of composites.

**Table 1 molecules-28-01023-t001:** EVA/LLDPE 118W/120 phr flame-retardant (MH + RP) formulations as a function of RP (without ZB) content.

Content/Property	Ingredients	AP-587	AP-662	AP-663	AP-664	AP-665
Matrix polymers	EVA (%)	80	80	80	80	80
	LLDPE 118W (%)	20	20	20	20	20
Flame retardants	MH (phr)	120	119	118	117	116
	RP (phr)	-	1	2	3	4
Coloring agent	Carbon black (phr)	6.0	6.0	6.0	6.0	6.0
Antioxidant	Naugard Q (phr)	1.5	1.5	1.5	1.5	1.5
Room temperature	Tensile strength (MPa)	12.0 ± 0.3	12.5 ± 0.2	12.1 ± 0.1	12.1 ± 0.2	12.1 ± 0.0
	Elongation at break (%)	160 ± 16	175 ± 7	162 ± 13	162 ± 9	159 ± 9
LOI (%)		38.5	42.5	43.5	45.5	45.5
UL-94 test		-	V-2	V-2	V-2	V-1
Volume resistivity (Ωcm)		-	1.6 × 10^15^	1.6 × 10^15^	1.5 × 10^15^	1.6 × 10^15^

**Table 2 molecules-28-01023-t002:** EVA/EXACT 8201/CLNA-8400/EBM/120 phr flame-retardant (MH+ZB) formulations as a function of additional RP content.

Content/Property	Ingredients	AP-651	AP-670	AP-671	AP-672	AP-673
	EVA (%)	75	75	75	75	75
Matrix polymers	EXACT 8201 (%)	10	10	10	10	10
	CLNA-8400 (%)	6	6	6	6	6
	EBM (%)	9	9	9	9	9
	MH (phr)	120	114	114	114	114
Flame retardants	RP (phr)	-	2	3	4	5
	ZB (phr)	-	6	6	6	6
Coloring agent	Carbon black (phr)	6.0	6.0	6.0	6.0	6.0
Antioxidant	Naugard Q (phr)	1.5	1.5	1.5	1.5	1.5
Room temperature	Tensile strength (MPa)	13.7 ± 0.2	13.6 ± 0.2	13.3 ± 0.2	12.9 ± 0.4	12.8 ± 0.4
	Elongation at break (%)	228 ± 3	194 ± 10	198 ± 7	187 ± 15	194 ± 4
Thermal aging at 100 °C for 168 h	Tensile strength (MPa)	13.5 ± 0.1	13.6 ± 0.8	13.0 ± 0.3	13.6 ± 0.3	13.4 ± 0.2
	Retention of tensile strength (%)	98.5	100	97.7	105.4	104.7
	Elongation at break (%)	208 ± 9	186 ± 7	182 ± 6	180 ± 7	176 ± 9
	Retention of elongation at break (%)	91.2	95.9	91.9	96.3	90.7
LOI (%)		29.5	31.5	32.5	33.5	34.5
UL-94 test		-	V-0	V-0	V-0	V-0
Volume resistivity (Ωcm)		-	7.2 × 10^13^	7.3 × 10^13^	7.2 × 10^13^	7.3 × 10^13^
Cone calorimeter	TTI (s)	28	NP	27	NP	28
	PHRR (KW/m^2^)	243		275		214
	FPI (m^2^s/KW)	0.115		0.098		0.131

NP: not performed.

**Table 3 molecules-28-01023-t003:** EVA/Tafmer DF805/CLNA-8400/EBM/120 phr flame-retardant (MH+ZB) formulations as a function of additional RB content.

Content/Property	Ingredients	AP-656	AP-674	AP-675	AP-676	AP-677
	EVA (%)	70	70	70	70	70
Matrix polymers	Tafmer DF805 (%)	15	15	15	15	15
	CLNA-8400 (%)	6	6	6	6	6
	EBM (%)	9	9	9	9	9
	MH (phr)	120	114	114	114	114
Flame retardants	RP (phr)	-	2	3	4	5
	ZB (phr)	-	6	6	6	6
Coloring agent	Carbon black (phr)	6.0	6.0	6.0	6.0	6.0
Antioxidant	Naugard Q (phr)	1.5	1.5	1.5	1.5	1.5
Room temperature	Tensile strength (MPa)	13.9 ± 0.4	13.3 ± 0.1	13.5 ± 0.4	13.3 ± 0.0	13.3 ± 0.2
	Elongation at break (%)	222 ± 11	212 ± 8	211 ± 6	193 ± 17	208 ± 10
Thermal agingat 100 °C for 168 h	Tensile strength (MPa)	13.2 ± 0.1	13.4 ± 0.2	13.7 ± 0.3	13.6 ± 0.3	14.0 ± 0.4
	Retention of tensile strength (%)	95.0	100.7	101.5	102.3	105.3
	Elongation at break (%)	229 ± 9	187 ± 7	192 ± 3	190 ± 3	183 ± 7
	Retention of elongation at break (%)	103.2	88.2	91.0	98.4	88.0
LOI (%)		30.0	32.0	32.5	33.0	34.0
UL-94 test			V-0	V-0	V-0	V-0
Volume resistivity (Ωcm)			7.4 × 10^13^	8.0 × 10^13^	9.1 × 10^13^	8.4 × 10^13^
Cone calorimeter	TTI (s)	29	NP	28	NP	27
	PHRR (KW/m^2^)	291		224		215
	FPI (m^2^s/KW)	0.100		0.125		0.126

NP: not performed.

**Table 4 molecules-28-01023-t004:** EVA/EXACT 8201/CLNA-8400/125 phr flame-retardant (MH+ZB+RP) formulations as a function of EBM content (First).

Content/Property	Ingredients	AP-673	AP-686	AP-687	AP-688	AP-689
	EVA (%)	75	75	75	75	75
Matrix polymers	EXACT 8201 (%)	10	10	10	10	10
	CLNA-8400 (%)	6	8	10	12	14
	EBM (%)	9	7	5	3	1
	MH (phr)	114	114	114	114	114
Flame retardants	RP (phr)	5	5	5	5	5
	ZB (phr)	6	6	6	6	6
Coloring agent	Carbon black (phr)	6.0	6.0	6.0	6.0	6.0
Antioxidant	Naugard Q (phr)	1.5	1.5	1.5	1.5	1.5
Room temperature	Tensile strength (MPa)	12.8 ± 0.4	12.6 ± 0.5	11.6 ± 0.5	11.4 ± 0.7	11.4 ± 0.3
	Elongation at break (%)	194 ± 4	184 ± 7	186 ± 5	163 ± 17	152 ± 11
	Tensile strength (MPa)	13.4 ± 0.2	12.5 ± 0.2	11.9 ± 0.1	11.5 ± 0.2	11.4 ± 0.2
Thermal aging at 100 °C for 168 h	Retention of tensile strength (%)	105	99	103	101	100
	Elongation at break (%)	176 ± 9	183 ± 4	168 ± 15	173 ± 8	159 ± 10
	Retention of elongation at break (%)	91	99	90	106	105
LOI (%)		34.5	35.5	35.5	36.0	37.5
UL-94 test		V-0	V-0	V-0	V-0	V-0
Volume resistivity (Ωcm)		7.3 × 10^13^	7.3 × 10^13^	1.6 × 10^14^	3.1 × 10^14^	1.0 × 10^15^
	TTI (s)	28	NP	27	NP	26
Cone calorimeter	PHRR (KW/m^2^)	214		216		265
	FPI (m^2^s/KW)	0.131		0.125		0.098

NP: not performed.

**Table 5 molecules-28-01023-t005:** EVA/EXACT 8201/CLNA-8400/125 phr flame-retardant (MH+ZB+RP) formulations as a function of EBM content (Second).

Content/Property	Ingredients	AP-730	AP-731	AP-732	AP-733	AP-734
	EVA (%)	75	75	75	75	75
Matrix polymers	EXACT 8201 (%)	10	10	10	10	10
	CLNA-8400 (%)	13	11	9	7	6
	EBM (%)	2	4	6	8	9
	MH (phr)	114	114	114	114	114
Flame retardants	RP (phr)	5	5	5	5	5
	ZB (phr)	6	6	6	6	6
Coloring agent	Carbon black (phr)	6.0	6.0	6.0	6.0	6.0
Antioxidant	Naugard Q (phr)	1.5	1.5	1.5	1.5	1.5
Room temperature	Tensile strength (MPa)	12.0 ± 0.1	12.3 ± 0.1	12.5 ± 0.2	13.3 ± 0.2	12.8 ± 0.4
	Elongation at break (%)	170 ± 6	186 ± 14	184 ± 4	188 ± 7	194 ± 4
Thermal aging at 100 °C for 168 h	Tensile strength (MPa)	10.5 ± 0.1	10.7 ± 0.3	10.9 ± 0.3	11.5 ± 0.2	13.4 ± 0.2
	Retention of tensile strength (%)	87	87	87	86	105
	Elongation at break (%)	164 ± 2	187 ± 9	183 ± 10	196 ± 4	176 ± 9
	Retention of elongation at break (%)	96	100	99	104	91
LOI (%)		36.0	35.0	35.0	35.0	34.5
UL-94 test		V-0	V-0	V-0	V-0	V-0
Volume resistivity (Ωcm)		8.3 × 10^14^	1.6 × 10^14^	7.9 × 10^13^	6.6 × 10^13^	7.3 × 10^13^

## Data Availability

There are no linked research datasets for this submission. Data will be made available on request.

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
