# Peer review of "A Systematic Investigation on the Influence of Intumescent Flame Retardants on the Properties of Ethylene Vinyl Acetate (EVA)/Liner Low Density Polyethylene (LLDPE) Blends"

_molecules, 2023, doi:10.3390/molecules28031023_

Round 1

Reviewer 1 Report

This paper deals with an investiagtion of different flame retardants for EVA/LLDPE blends. I general this is a very interesting topic in terms of application in the field of power cables.

Nevertheless, the publication is quite hard to follow because of a variety of materials and mixing of results and state of the art. More scientific depth would be desirable. I recommend a major revision before publishing this article.

Detailed comments:

Introduction:

- The introduction gives good basics on EVA/PE blends and their applications as well as intumescent flame retardants in general. But information is missing on the state of the art of flame retardants in EVA/PE blends (some information is given in the results part, but should be explained in the introduction)

Experimental Section:

- Testing speed of 500 mm/min (line 108) seems very high. Is the number correct?

Results and discussion

- In every chapter the compositions of your material are completely changing, for example:

EVA: 3.1: 80% - 3.2: 75% - 3.3.: 70%

Total amount of flame retardants: 3.1: 120phr - 3.2: 120-125 phr

By changing more than one component from one step to the other it is hard to follow for the reader (also the tables are quite complex). It would be nice, if you make your thoughts more clear in the text. A graphical abstract or table, where one can see the changes between the chapters in a easy way could also help.

- Units (wt% / vol% / phr) are missing in the tables.

- Especially in chapter 3.1 you mix up your own results with state of the art results. You can do some correlations, but keep most of the state of the art in the introduction.

- 3.2: State why you change from LLDPE 118 W to EXACT+CLNA

- Figure 5: AP671 has the lowest overall FR content (117 instead of 120) - can be a reason for the effect you see

- In general the interaction between the different components should be discussed more in detail. You have 11 different materials that can change you morphology, crystallinty... drastically, and therefore the product properties. Some insights into the materials structure (e.g. SEM) would be helpful.

Reviewer 2 Report

In this paper, RP was used to investigate the effect on the flame retardancy, mechanical properties of EVA/LLDPE composites in detail. The comments are as follows:

1. Line 53, please revise MH to magnesium hydroxide because it is the first appearing in the introduction, please revise others, such EVA, RP, ZB, and so on.

2. Please add some references about the flame retardancy of EVA/LLDPE in the introduction part.

3. What about the mechanical properties of EVA/LLDPE with flame retardants?

4. Lines 229-231, how do flame retardants enhance the material both chemically and physically?

5. There are too many Figures, and some of Figures can be combined together, for example, Figures 19-22 can be combined to one Figure.

6. English needs improving, such as Line 161 “This implies that the change is because of RP rather than ZB because”.

Reviewer 3 Report

The manuscript is a study about the improving of mechanical and flame retardant properties of ethylene- vinyl acetate composites using red phosphorus, zinc borate and a terpolymer (‘ethylene-butyl acrylate-maleic anhydride’). Flame retardants as magnesium hydroxide and additional reagents were also involved. The obtained data are compared with other similar materials/approaches. The applicability of the obtained composite is in the field of materials that are used to sheath the electric power cable.

Reviewer comments/suggestions:

-More details about strategy of the approach used in the developing of the presented study (that are reflected also even in the chapters’ names and in the used measure units-“phr: parts per hundred resin/rubber”) accompanied with proper referinces are useful. E.g. how were chosen the blanks related to the main product and base/raw materials (i.e. a “blank” is the main composite without one or more of its components); how was chosen the variation of the components amount?

-Some of the involved raw materials (Chapter 2.1) are improper presented (without purity, concentration, features, etc.).

-The preparation method (Chapter 2.2) is incompletely. The addition/entry (concentration, amount, volume, times, etc.) data of the components that are presented in final composites are missing. The references of the patents are not enough.

-A schematic presentation of the preparation process is useful.

-The meaning/name of terms as “EVA/LLDPE 118W/120phr flame retardants (MH+RP)” is not explained.

-Lines 138-142. Stated information is not clear (the relevance/importance of it and the acting difference between MH and HH);

-Information about terms as AP-587, AP-662, AP-663, AP-664, AP-665, etc. with relevant references is not stated.

-The figures present a low graphical quality.

- In the Figures 11 and 17, for comparison the sample images before cone calorimeter tests are useful.

Round 2

Reviewer 2 Report

This paper can be accepted in present form.